# In Vitro Investigations on Optimizing and Nebulization of IVT-mRNA Formulations for Potential Pulmonary-Based Alpha-1-Antitrypsin Deficiency Treatment [note 1]

**DOI:** 10.3390/pharmaceutics13081281

**Published:** 2021-08-17

**Authors:** Shan Guan, Max Darmstädter, Chuanfei Xu, Joseph Rosenecker

**Affiliations:** 1National Engineering Research Center of Immunological Products, Department of Microbiology and Biochemical Pharmacy, Third Military Medical University, Chongqing 400038, China; xu1982978434@163.com; 2Department of Pediatrics, Ludwig-Maximilians University of Munich, 80337 Munich, Germany; maxdarmstaedter@gmail.com

**Keywords:** gene therapy, IVT-mRNA, alpha-1-antitrypsin deficiency, nebulization, pulmonary delivery

## Abstract

In vitro-transcribed (IVT) mRNA has come into focus in recent years as a potential therapeutic approach for the treatment of genetic diseases. The nebulized formulations of IVT-mRNA-encoding alpha-1-antitrypsin (A1AT-mRNA) would be a highly acceptable and tolerable remedy for the protein replacement therapy for alpha-1-antitrypsin deficiency in the future. Here we show that lipoplexes containing A1AT-mRNA prepared in optimum conditions could successfully transfect human bronchial epithelial cells without significant toxicity. A reduction in transfection efficiency was observed for aerosolized lipoplexes that can be partially overcome by increasing the initial number of components. A1AT produced from cells transfected by nebulized A1AT-mRNA lipoplexes is functional and could successfully inhibit the enzyme activity of trypsin as well as elastase. Our data indicate that aerosolization of A1AT-mRNA therapy constitutes a potentially powerful means to transfect airway epithelial cells with the purpose of producing functional A1AT, while bringing along the unique advantages of IVT-mRNA.

## 1. Introduction

Nucleic acid-based therapeutics encoding specific proteins of interest have shown great potential in the treatment of devastating diseases such as genetic disorders, infectious diseases, cancer and cardiovascular diseases [1,2]. In vitro-transcribed messenger RNA (IVT-mRNA) has emerged as an alternative to the conventional DNA-based therapeutic and provides many unique features to be a promising drug candidate, for example, high efficiency in transfecting non-dividing cells and ease of production [3]. Most importantly, IVT-mRNA offers a huge advantage in terms of safety; it has no risk of insertional mutagenesis. IVT-mRNA is able to rapidly express the desired protein in the cytoplasm and automatically degrade afterwards, so the protein expression could be easily controlled [4,5]. Pioneering works initiated by Katalin Karikó et al. have enabled an in-depth understanding of the relationship between IVT-mRNA structure and its immunogenicity profile [6,7]. With the utilization of chemically modified nucleotides and advanced purification methods, the stability, immunogenicity and expression efficiency of IVT-mRNA have been greatly improved [8]. A large number of IVT-mRNA-based therapeutics is tested in clinical trials [9,10,11,12,13,14], and recently IVT-mRNA has displayed great potency in developing efficacious vaccine approaches to eliminate the spread of severe acute respiratory syndrome coronavirus 2 (SARS-CoV-2), the causative culprit of coronavirus disease 2019 (COVID-19) [15,16].

For preclinical and clinical applications, different routes of delivery of IVT-mRNA complexes to the target tissue are being tested, such as intramuscular, intradermal, intranodal, subcutaneous, intravenous and intrathecal. However, an important but not well-investigated field is the pulmonary delivery of IVT-mRNA. The lung airway, with its large surface area, represents an attractive target for IVT-mRNA-based gene therapy approaches designed to treat inherited monogenic diseases [17]. The local delivery of IVT-mRNA through the respiratory tract is a simple administration route that is deprived of drawbacks inherent to intravenous administration. The lung has a rich capillary network and strong angiogenic capacity, which can mediate secreted proteins into the circulatory system [18,19]. Because the inhalation of aerosols is a highly acceptable and tolerable route for the patient, the nebulized formulation tends to be more evenly distributed throughout the respiratory tract [20]. Nebulization is one of the most exploited methods for introducing gene vectors into the lung both in animal and clinical studies [21]. However, aerosolized gene therapy formulations can be inefficient: shearing force, preferential nebulization of the solute and adhesion to plastic can mean that as little as 10% of nucleic acid payload in the nebulization chamber is successfully emitted through the mouthpiece [22]. Nevertheless, optimized formulations and advanced nebulization strategies have significantly improved the situation; it has been proven by recent studies that nebulization of IVT-mRNA complexes is feasible, and could be successfully used for in vitro and in vivo applications [23,24]. As a result, nebulized IVT-mRNA formulations appears to be an attractive therapeutic approach for the treatment of a broad range of respiratory diseases. One of the most commonly quoted examples is alpha-1-antitrypsin deficiency [19].

Alpha-1-antitrypsin deficiency (AATD) is one of the most common hereditary disorders in Caucasians of European decent [25]. It is an autosomal recessive disorder caused by mutations within the SERPINA1 gene and characterized by low levels of alpha-1-antitrypsin (A1AT) in the serum [26,27]. A1AT is mainly synthesized and secreted by hepatocytes, but its primary function is to inhibit activity of neutrophil elastase in the lung, a serine protease that is able to destroy alveoli and cleave components of the extracellular matrix [28]. The lower respiratory tract of AATD patients is not protected against the destructive influence of neutrophil elastase, so AATD patients have a high risk of developing emphysema and chronic obstructive pulmonary disease [29,30]. Besides anti-protease properties, A1AT possesses multiple anti-inflammatory and tissue protective properties [31]. Recent clinical findings indicated that lower A1AT levels were related to worse prognosis in COVID-19 patients [32]. Follow-up investigations revealed that A1AT inhibits important proteases in the SARS-CoV-2 infection process [33]. To date, the only available treatment option for AATD is augmentation therapy being approved for selected patients with severe AATD-related pulmonary emphysema [34]. Nevertheless, A1AT augmentation therapy is costly and requires frequent intravenous infusion of A1AT purified from pooled human plasma, which has the risk of viral contamination and allergic reactions [35]. In contrast, IVT-mRNA-based gene therapy could eliminate the burden of protein infusion and significantly reduce the costs and associated risks [36]. A previous study suggested that A1AT expression could be observed in the lung and liver of mice which intravenously be injected by a single dose of IVT-mRNA-encoding A1AT (A1AT-mRNA) [37]. Because pathologies related to the deficiency of A1AT mainly concern the lung, aerosolized A1AT-mRNA could potentially be applicable for the treatment of AATD patients. However, to the best of our knowledge, there is no study published up to now which has explored the potential of nebulized IVT-mRNA therapy for the treatment of AATD.

For the delivery of IVT-mRNA-based therapeutics to the patient by inhalation, several challenges have to be addressed. One of the most important tasks would be protecting the vulnerable mRNA molecules against shear forces caused by nebulization. In this context, we carried out this proof-of-concept study aiming to investigate whether aerosolized IVT-mRNA-carrying formulations could successfully transfect airway epithelial cells and produce sufficient amounts of functional A1AT. In order to facilitate the efficient intracellular delivery of IVT-mRNA, Lipofectamine2000 was used as a transfection reagent due to the fact that it has shown favorable results in the context of IVT-mRNA-mediated in vitro transfection [23,38]. After establishing an optimal transfection protocol, we evaluated the transfection profile of complexes carrying A1AT-mRNA and confirmed the production of A1AT in transfected bronchial epithelial cells. The influence of the nebulization process towards the biological activity of IVT-mRNA was investigated in detail. We also developed an improved protocol to guarantee the transfection efficiency of nebulized IVT-mRNA lipoplexes. Since AATD is not caused by a lack of protein production but an inability to secrete functional A1AT protein, it is important to evaluate the functionality of the secreted A1AT from cells transfected by nebulized IVT-mRNA formulation. To that end, the function of secreted A1AT was tested in trypsin and elastase inhibition assay. Taken together, the data obtained in the current study indicate that nebulization of lipoplexes containing A1AT-mRNA is feasible for in vitro production of functional A1AT.

## 2. Materials and Methods

### 2.1. Materials

Chemically modified in vitro-transcribed messenger RNA-encoding Metridia luciferase (MetLuc-mRNA), green fluorescent protein (GFP-mRNA) and human alpha-1-antitrypsin (A1AT-mRNA) were generously provided by Ethris GmbH (Planegg, Germany). Lipofectamine2000 was obtained from Invitrogen (1 mg/mL, Darmstadt, Germany). 4,6-diamidino-2-phenylindole (DAPI) was purchased from Life technologies GmbH (Karlsruhe, Germany). 3-(4,5-dimethyl-2-tetrazolyl)-2,5-diphenyl-2H-tetrazolium bromide (MTT) solution was purchased from Roche^®^ Applied Science (Indianapolis, IN, USA). All the other reagents and solvents were of the highest purity commercially available.

### 2.2. Cell Culture

Human bronchial epithelial cell line, 16HBE14o- (16HBE), was generously provided by Prof. Dr. Dieter C. Gruenert (University of California at San Francisco, CA, USA). The cells were cultured in a 75 cm^2^ culture flask in Ham’s F-12K (Kaighn’s) Medium (Gibco, Life Technologies, Berlin, Germany) supplemented with 10 % of heat-inactivated fetal bovine serum (Gibco, Life Technologies) and 5 mL of penicillin/streptomycin (10,000 units/mL, Gibco, Life Technologies). Cells were incubated at 37 °C in an incubator (Heraeus Instruments GmbH, Hanau, Germany) in 5% CO_2_ atmosphere. The cells were split when they were 90% confluent. Unless specified, cells were pre-seeded in 24-well plates at a density of 7.5 × 10^4^ cells/well 24 h before the transfection experiments in order to reach a 70–80% confluence.

### 2.3. Preparation of IVT-mRNA Complexes

IVT-mRNA was formulated with Lipofectamine2000 in serum-free OptiMEM (Gibco, Life Technologies, Germany) or serum-free Ham’s F-12K medium (Gibco, Life Technologies, Germany), according to the manufacturer’s instructions at RT (room temperature). For example, 50 µL solution containing 6 µL IVT-mRNA (0.1 µg/µL) was gently mixed with 50 µL solution containing a certain amount of Lipofectamine2000 (e.g., 2.4 µL, 3.6 µL, 4.8 µL). The resulting formulation was incubated at 25 °C for 10 min prior to further use. Aliquots of the final solution were added to the cells.

### 2.4. Size Measurements

The IVT-mRNA/Lipofectamine2000 complexes were prepared in two settings using the above-described methods at RT. First setting: 1.2 µL, 2.4 µL and 3.6 µL of Lipofectamine2000 in 50 µL OptiMEM were mixed with 2 µL, 4 µL and 6 µL MetLuc-mRNA in 50 µL OptiMEM, respectively; second setting: 2.4 µL, 3.6 µL and 4.8 µL of Lipofectamine2000 in 50 µL OptiMEM was incubated with 6 µL of MetLuc-mRNA in 50 µL OptiMEM. A total amount of 900 μL of OptiMEM was added to each sample after the incubation. The particle sizes of these complexes were determined with a Zeta-Sizer (Brookhaven Instruments, Long Island, NY, USA) at 25 °C. For the measurement, 1 mL of IVT-mRNA solution was pipetted into a cuvette.

### 2.5. Transfection of Cultured Cells

If not specified otherwise, transfection studies were performed in 24-well plates. After the removal of growth medium, cells were rinsed with PBS (Gibco Life Technologies, Berlin, Germany). A total of 450 µL of serum-free OptiMEM or serum-free Ham’s F-12K medium was added per well, and 50 µL IVT-mRNA complexes (prepared as described above) were subsequently added in replicates of four or more. The complexes were incubated with the cells for 2 h at 37 °C in a humidified 5% CO_2_-enriched atmosphere; the transfection medium was replaced with 1 mL fresh culture medium supplemented with 10% FBS and 1% (*v*/*v*) penicillin/streptomycin. Transfection aliquots of supernatants were collected at predetermined time points. The cell culture medium was replaced with fresh one after each sampling. Naked IVT-mRNA (IVT-mRNA formulation prepared without using delivery systems or transfection reagents) was used as a negative control.

### 2.6. Luciferase Assay

The MetLuc-mRNA/Lipofectamine2000 complexes were prepared in two schemes: (1) 2.4 µL, 3.6 µL or 4.8 µL Lipofectamine2000 in 50 µL solutions were mixed with 6 µL MetLuc-mRNA (0.1 µg/µL) in 50 µL solutions; (2) 1.2 µL, 2.4 µL or 3.6 µL Lipofectamine2000 were mixed, respectively, with 2 µL, 4 µL or 6 µL MetLuc-mRNA (0.1 µg/µL) with a final volume of 100 µL. 16HBE cells were incubated with above MetLuc-mRNA/Lipofectamine2000 complexes at the volume of 50 µL/well for 2 h. The transfection efficiency was evaluated 24 h after adding complexes to the cells. A total of 50 µL of supernatant from each sample was added to a 96-well plate. This was followed by the addition of 40 µL of the Metridia luciferase substrate (coelenterazine, InvivoGen, Toulouse, France). The emitted light was measured with a microplate reader (FLUOstar Optima, BMG Labtech, Ortenberg, Germany) and its activity is expressed in relative light units.

### 2.7. Fluorescence Microscopy

For the transfection of 16HBE cells with IVT-mRNA-encoding GFP, complexes were prepared as described above. Totals of 2.4 µL or 3.6 µL Lipofectamine2000 in 50 µL medium were mixed, respectively, with 4 µL or 6 µL GFP-mRNA (0.1 µg/µL) in serum-free OptiMEM or serum-free Ham’s F-12K medium to prepare complexes for transfection. To visualize GFP-mRNA-transfected cells, 16HBE cells were seeded at a density of 15.0 × 10^4^ cells/well in IBIDI 8-well slides (IBIDI GmbH, Gräfelfing, Germany) 24 h before transfection to reach a monolayer (~100% confluence). The complexes were incubated with the cells for 2 h in an incubator. The transfection efficiency was evaluated 24 h after transfection using fluorescence microscopy (Zeiss Axiovert 200 M, Carl Zeiss Microscopy GmbH, Munich, Germany). Quantitative analysis on the ratios of successfully transfected cells in the captured images was performed by ImageJ (version 1.53j, NIH, Bethesda, MD, USA).

### 2.8. MTT-Based Cytotoxicity Assay

16HBE cells were plated into a 96-well plate (3.0 × 10^4^ cells/well) 24 h before transfection with IVT-mRNA lipoplexes (~80% confluence). After 2 h of incubation, the complexes were removed and 100 µL of fresh medium was added. Cell viability was measured 24 h after transfection. To that end, 10 µL of the MTT solution was added to the cells and incubated for 2 h. Subsequently, 100 µL of the solubilization solution (10% SDS in 0.01M HCL, Roche^®^ Applied Science, Indianapolis, IN, USA) was added. After 24 h, the absorbance was measured with a microplate reader (FLUOstar Optima, BMG Labtech) at 600 nm with a reference above 650 nm. Untreated cells were used as controls. The cell viability was calculated as a relative value (in percentage) compared to the control group.

### 2.9. Nebulization

For nebulization experiments, we used an improved transfection protocol to ensure a higher transfection efficiency after being aerosolized. The IVT-mRNA/Lipofectamine2000 complexes were prepared by mixing 18 µL MetLuc-mRNA or A1AT-mRNA with 7.2 µL, 10.8 µL and 14.4 µL Lipofectamine2000 in serum-free OptiMEM or serum-free Ham’s F-12K medium with a total volume of 100 µL. The above solutions were incubated for 10 min at RT. This was followed by the addition of 2900 µL medium. The solution was divided into two fractions. A small fraction was kept apart and was used as a non-nebulized control. The other part was pipetted into the PARI Boy^®^ Jet-Nebulizer (Pari GmbH, Starnberg, Germany) and was aerosolized for 5 min; the aerosolized solution was collected in a micro-centrifuge tube (Eppendorf™ PCR Clean, Thermo Fisher Scientific, Berlin, Germany) and used as the “nebulized” group. Subsequently, both fractions (20 µL/well) were pipetted onto a 96-well plate, in which 16HBE cells were pre-seeded at a density of 3.0 × 10^4^ cells/well one day before (~80% confluence). The complexes were incubated with the cells for 2 h in presence of 80 µL/well serum-free OptiMEM. After their removal, 0.1 mL of fresh culture medium was added. The luciferase activity or the A1AT related assay was measured 24 h after transfection.

### 2.10. Detection of Alpha-1-Antitrypsin Using Enzyme-Linked Immunosorbent Assay (ELISA)

For the detection of alpha-1-antitrypsin (A1AT), we used an alpha-1-antitrypsin human ELISA kit (Abcam, Cambridge, UK). In this assay, a 96-well plate is coated with a capture antibody, which is specifically used for the recognition of alpha-1-antitrypsin. Cell supernatants were collected 24 h after transfecting 16HBE cells with A1AT-mRNA/Lipofectamine2000. Untreated samples were used as blank controls, Lipofectamine2000-treated samples were served as mock controls and MetLuc-mRNA/Lipofectamine2000-treated counterparts were applied as negative controls (NC). Supernatant was centrifuged at 3000× *g* for 10 min to remove cell debris. Subsequently, 10 µL of supernatants were diluted in medium. A total of 50 µL of standard solutions or samples was pipetted onto the microplate. The wells were covered with a sealing tape and incubated for 2 h. Then, 50 µL of a biotinylated alpha-1-antitrypsin antibody (Abcam, Cambridge, UK) were added. After 1 h of incubation, 50 µL of the streptavidin-peroxidase conjugate were added. After each step, the plate was washed 5 times with a mild detergent. Subsequently, a chromogen substrate and, finally, a stop solution were added. The absorbance was measured with a plate reader (FLUOstar Optima, BMG Labtech, Ortenberg, Germany) at 405 nm.

### 2.11. Alpha-1-Antitrypsin Functional Assay

Trypsin degradation: 5 µL of trypsin solution (0.02 U/µL, Abcam, UK) were added to 45 µL trypsin assay buffer. A total of 50 µL supernatants of A1AT-mRNA/Lipofectamine2000-transfected cells was added to this solution. After 10 min of incubation, 50 µL trypsin substrate solution (Na-Benzoyl-DL-arginine-b-naphthylamide hydrochloride, Abcam, UK) were added. The solution was mixed by vortexing. The colorimetric reaction was followed by measuring fluorescence at 405 nm with a plate reader (FLUOstar Optima, BMG Labtech, Ortenberg, Germany).

Elastase degradation: to perform this assay, 5 µL of the elastase solution (0.1 U/µL, from EnzChek Elastase Assay Kit, Molecular Probes, Life Technologies, Germany) were added to 45 µL of the reaction buffer (1 M Tris-HCl, pH 8, containing 2 mM sodium azide). Afterwards, 50 µL of supernatant from cells transfected with A1AT-mRNA/Lipofectamine2000 were added and incubated at 37 °C for 10 min. The reactions were diluted in 400 µL reaction buffer containing the chromogenic substrate (160 nmol, *N*-Methoxysuccinyl-Ala-Ala-Pro-Val-*p*-nitroanilide, MeO-SucAAPV-pNA, Sigma). The colorimetric reaction was evaluated by measuring the absorbance of 200 µL reaction mixtures at 410 nm with a plate reader (FLUOstar Optima, BMG Labtech, Ortenberg, Germany).

For both assays, untreated samples were used as blank controls and their absorbance were set as 100% (whose inhibition activity was 0% correspondingly). Lipofectamine2000-treated samples served as mock controls and MetLuc-mRNA/Lipofectamine2000-treated counterparts were adopted as negative controls (NC).

### 2.12. Immunofluorescence

16HBE cells were transfected with IVT-mRNA-encoding A1AT, as described above. IVT-mRNA lipoplexes were removed after 2 h of incubation. Fresh cell culture medium and Brefeldin-A were added. After 24 h, cells were washed 3 times with PBS. To fixate the cells in their current state, 1 mL of a fixation buffer (4% paraformaldehyde in PBS, BioLegend, San Diego, CA, USA) was used. After an incubation period of 10 min, the cells were washed with PBS again. Subsequently, 1 mL of fix/perm buffer (BD Biosciences, Heidelberg, Germany) was added. The cells were incubated with the buffer for 15 min. Afterwards, a human A1AT-specific antibody (NBP1-90309, Novus biologicals, Centennial, CO, USA) was added. To ensure bonding between the antigen of interest and the detection antibody, the incubation time was 1 h at room temperature (RT). The cells were washed again with PBS to remove an excess of antibodies. To keep the cells permeabilized, we treated them again with a perm/wash buffer. Subsequently, a Goat anti-Rabbit IgG ReadyProbes™ secondary antibody flagged with AlexaFluor^®^594 (Life Technologies, Germany) was added. Debris was removed by washing the cells with PBS. For intracellular orientation, 300 µL DAPI (300 nM in PBS) was used to stain the nucleus. Two controls were used, one using a non-specific antibody, as well as the secondary antibody, and for the second, only the secondary antibody was used.

### 2.13. Statistical Analysis

Data for all bar charts were prepared using means and error bars that correspond to standard deviations. Statistical analysis was performed using Prism 8 (GraphPad Software Inc, San Diego, CA, USA). An ANOVA followed by Dunnett’s test were applied for comparisons between different groups. The statistical significance of differences between two groups was analyzed by two-tailed Student’s *t*-tests, and differences were considered statistically significant when *p* < 0.05.

## 3. Results

### 3.1. Optimization of the Transfection Process

#### 3.1.1. Finding the Optimal Transfection Conditions

The IVT-mRNA/Lipofectamine2000 complexes were prepared in OptiMEM, a medium specially designed for transfection. Dynamic light scattering measurements revealed that the size of IVT-mRNA/Lipofectamine2000 complexes prepared under different conditions and settings were typically a few hundred nanometers, as depicted in Table 1 (non-nebulized). In order to evaluate the influence of the components’ ratio on the transfection efficiency of IVT-mRNA lipoplexes, increased amounts of Lipofectamine2000 were formulated with a fixed amount of IVT-mRNA. The IVT-mRNA-encoding secreted Metridia luciferase (MetLuc-mRNA) was selected as a reporter system to transfect human bronchial epithelial cells [16HBE14o- (16HBE)]. As shown in Figure 1A, there was no significant difference in the luciferase activities between the studied conditions, indicating that complexes prepared by mixing Lipofectamine2000 with IVT-mRNA at these ratios transfect 16HBE cells with almost equal efficiency.

Subsequently, 2 µL, 4 µL or 6 µL of MetLuc-mRNA were mixed with Lipofectamine2000 at the volume ratio of 1:0.6 to reveal whether increased amount of IVT-mRNA could translate into higher transfection efficiencies. The results showed that lipoplexes containing different amounts of MetLuc-mRNA, but prepared and incubated with the same medium, resulted in a similar transfection profile (Figure 1B). To determine the influence of different media on the transfection efficiency, the complexes were first prepared in OptiMEM or the medium routinely used for growing 16HBE cells (Ham’s F-12K); both cases were serum-free throughout the study. The transfection efficiency was evaluated 24 h after adding complexes to the cells by measuring the luciferase activity. When MetLuc-mRNA complexes were prepared and incubated with 16HBE cells in Ham’s F-12K medium, the levels of luciferase activity were significantly lower than counterparts prepared by OptiMEM (Figure 1B).

In the next set of experiments, we employed other solutions that are commonly used in a clinical setting to prepare IVT-mRNA complexes. To this end, we transfected 16HBE cells with complexes prepared in glucose, sucrose or sodium chloride and evaluated their transfection efficiencies. To our surprise, luciferase activity mediated by IVT-mRNA/Lipofectamine2000 complexes prepared in a 5% glucose or 5% sucrose solution was hardly measurable. Maximal levels of luciferase activity obtained were 87,963 and 25,240 RLU for 5% glucose and 5% sucrose, respectively. On the other hand, lipoplexes prepared in saline were relatively more efficient, as demonstrated in Figure 1C. Maximal levels of luciferase activity obtained were above 1,000,000 RLU. However, it is worth to note that the transfection efficiency mediated by complexes prepared in saline was still low compared to the case of OptiMEM.

In order to improve the transfection efficiency that was mediated by glucose or sucrose solution, we first prepared MetLuc-mRNA/Lipofectamine2000 lipoplexes using 0.9% sodium chloride solution, and subsequently incubated them with 16HBE cells in 5% glucose or 5% sucrose. As shown in Figure 1D, the transfection efficiencies of these lipoplexes showed some increase compared with those lipoplexes prepared in 5% glucose or 5% sucrose. However, the level of transfection was lower than those prepared and incubated in 0.9% sodium chloride solution, suggesting the sodium chloride ions are relatively advantageous in facilitating the transfection of the lipoplexes. Based on all the above results, we chose OptiMEM and Ham’s F-12K medium to prepare IVT-mRNA/Lipofectamine2000 lipoplexes and incubate the formulation with 16HBE cells in the following studies.

#### 3.1.2. Duration of Protein Production

IVT-mRNA is considered to be a relatively unstable molecule compared with other types of nucleic acids. Transfection of cultured cells mediated by cationic lipid-based IVT-mRNA complexes is expected to be transient. To verify, 16HBE cells transfected with MetLuc-mRNA complexed with Lipofectamine2000 were monitored for luciferase production over a period of several days. Three Lipofectamine2000-to-MetLuc-mRNA ratios were tested. To ensure the maximal level of protein production, the MetLuc-mRNA/Lipofectamine2000 complexes were prepared using OptiMEM. As shown in Figure 2, significant levels of luciferase activity could be detected for four days. The highest levels of protein production were found 24 h after adding complexes to the cells.

#### 3.1.3. Evaluating Transfection Efficiency via GFP-mRNA

In addition to evaluating total levels of protein production in 16HBE cells, we also assessed numbers of transfected cells using green fluorescent protein as a reporter. To that end, we employed IVT-mRNA-encoding green fluorescent protein (GFP-mRNA) to prepare complexes with Lipofectamine2000. The transfection efficiency was evaluated 24 h after adding complexes to the cells by visualizing transfected cells via a fluorescent microscope (Figure 3A,B). Consistent to the result obtained from luciferase-mRNA-based transfection, a large number of cells transfected by GFP-mRNA/Lipofectamine2000 lipoplexes prepared in OptiMEM showed a positive signal for GFP, while relatively less GFP-positive cells were observed in the group transfected by GFP-mRNA lipoplexes prepared in Ham’s F-12K medium (Figure 3C).

### 3.2. Transfection with IVT-mRNA-Encoding Alpha-1-Antitrypsin (A1AT)

#### 3.2.1. A1AT Expression Mediated by A1AT-mRNA Lipoplexes

We then evaluated the transfection efficiency of complexes prepared by mixing Lipofectamine2000 with IVT-mRNA-encoding alpha-1-antitrypsin (A1AT-mRNA). To confirm the expression of A1AT, we stained the produced protein inside 16HBE cells with an antibody labelling method. This technique is based on indirect immunohistochemical staining using lactone antibiotic Brefeldin-A to inhibit transfected cells from secreting the produced A1AT. The inhibited cells are not able to secrete proteins via vesicles. After blocking the secretion of proteins, intracellular A1AT was labelled by a specific antibody and was visualized via a secondary antibody tagged with red fluorescent dye. A strong red signal could be observed within cells transfected by A1AT-mRNA/Lipofectamine2000 complexes, as shown in Figure 4A,B, while none of the red signal could be detected in untreated cells (Figure 4C), as well as in transfected cells that were treated with an unspecific primary antibody or secondary antibody (data not shown).

To further detect the amount of secreted A1AT in transfected 16HBE cells, an enzyme-linked immunosorbent assay (ELISA) was performed. Appendix A shows a representative ELISA microplate. The amounts of A1AT in samples transfected by A1AT-mRNA/Lipofectamine2000 complexes were calculated using a standard curve method. Non-treated samples were used as blank controls, and samples treated with Lipofectamine2000 and irrelevant IVT-mRNA-based counterparts (MetLuc-mRNA/Lipofectamine2000) served as mock controls and negative controls (NC), respectively. The results are presented in Figure 5. A1AT could not be detected from blank, mock or NC samples. Cells transfected with A1AT-mRNA-based lipoplexes prepared in OptiMEM showed a slightly higher rate of secreted A1AT. The maximal levels of A1AT were secreted by cells transfected with the lipoplexes prepared at the highest Lipofectamine2000-to-A1AT-mRNA ratios. However, there is no significant differences between “OptiMEM” group and “Ham’s F-12K” counterpart, as revealed by statistical analysis.

#### 3.2.2. Cell Viability after Transfection with Different IVT-mRNA Lipoplexes

An MTT assay was employed to assess the impact of lipoplexes carrying different types of IVT-mRNA (i.e., MetLuc-mRNA, GFP-mRNA and A1AT-mRNA) on the viability of 16HBE cells (Appendix A). OptiMEM or Ham’s F-12K was used to prepare IVT-mRNA complexes. Toxicity was evaluated 24 h after transfection, which is the time when the transfected cells need to produce maximal levels of the protein of interest and are required to deal with possible degraded products. Toxicity induced by formulations prepared in OptiMEM showed mild toxicities at the highest Lipofectamine2000-to-MetLuc-mRNA ratios (Appendix A), while none of the tested GFP-mRNA/Lipoplexes (Appendix A) and A1AT-mRNA/Lipoplexes (Appendix A) displayed significant toxicity in 16HBE cells.

### 3.3. Nebulization of IVT-mRNA Complexes

#### 3.3.1. Particle Size and Transfection Efficiency of Lipoplexes after the Nebulization

In order to confirm whether IVT-mRNA lipoplexes could tolerate the nebulization process, we compared the change in particle size of MetLuc-mRNA/Lipofectamine2000 complexes before and after nebulization, then evaluated the luciferase activity induced by nebulized MetLuc-mRNA/Lipofectamine2000 complexes with their non-nebulized counterparts. After nebulization, an increase in the particle size can be observed in all the settings (Table 1, “Nebulized”). Afterwards, we transfected 16HBE cells with nebulized MetLuc-mRNA complexes and non-nebulized control that was prepared in different media and with different Lipofectamine2000-to-IVT-mRNA ratios in order to evaluate the impact of the nebulization process on transfection efficiency. As shown in Figure 6, all groups of the nebulized MetLuc-mRNA/Lipofectamine2000 complexes that were prepared with the standard protocol (Protocol 1) were not as efficient in transfecting 16HBE cells as their non-nebulized counterparts. A significant decrease in transfection efficiency was detected in all nebulized lipoplexes prepared by protocol 1, regardless of the charge ratios or the medium that applied (Figure 6, Protocol 1), implying that Lipofectamine2000 could not protect IVT-mRNA against the shear force induced by the nebulizer. As a result, we needed to develop an improved protocol in which higher transfection efficiencies of the nebulized formulations could be reached. To this end, we kept the same Lipofectamine2000-to-IVT-mRNA ratio but used triple amounts of IVT-mRNA to prepare the lipoplexes (Protocol 2). The size of these lipoplexes was further increased due to the enlarged number of components, as well as nebulization process (Appendix A). The transfection efficiency of MetLuc-mRNA lipoplexes prepared by protocol 2 and their nebulized counterpart was evaluated; the results are shown in Figure 6. The nebulized lipoplexes prepared in OptiMEM using protocol 2 showed a significantly higher efficiency than the nebulized complexes prepared according to protocol 1 in all conditions (Figure 6A). When complexes were prepared in Ham’s F-12K medium, the differences between the standard transfection (protocol 1) and the improved nebulization protocol (protocol 2) were significant in some conditions, as shown in Figure 6B.

#### 3.3.2. Cell Viability following Transfection with Protocol 2

After obtaining sufficient transfection efficiency of the nebulized IVT-mRNA/Lipofectamine2000 complexes, it was crucial to evaluate their toxicity on 16HBE cells. The MTT-assay was performed 24 h after transfection with non-nebulized or nebulized lipoplexes, using protocol 2. As demonstrated in Figure 7, both non-nebulized and nebulized lipoplexes prepared in OptiMEM showed considerably enhanced cytotoxicity towards 16HBE cells in all three conditions. The enhanced toxicity may result from the increased concentration of Lipofectamine2000 and IVT-mRNA. It is worth to note that the diminished cytotoxicity of nebulized lipoplexes probably was correlated to the decreased transfection efficiency, indicating the nebulization process destroyed the formulation to a certain degree. Similarly, 16HBE cells incubated with IVT-mRNA lipoplexes prepared in Ham’s F-12K medium also showed significantly reduced viability, as displayed in Appendix A.

#### 3.3.3. Nebulization of A1AT-mRNA/Lipofectamine2000 Complexes

The main focus of the current study was to investigate the transfection efficiency of nebulized lipoplexes formulations containing A1AT-mRNA, and to confirm that the secreted A1AT protein was still functional. We transfected 16HBE cells with nebulized A1AT-mRNA/Lipofectamine2000 complexes prepared by protocol 2, as described above, and non-nebulized control in order to evaluate the impact of the nebulization process on the transfection efficiency. Non-treated, mock (only Lipofectamine2000)-treated and irrelevant counterpart (NC)-treated samples were added to ensure the measured protein all originated from successfully transfected A1AT-mRNA. ELISA was employed to assess the amount of secreted A1AT. As shown in Figure 8, the most abundant A1AT production was observed in the non-nebulized complexes prepared by mixing Lipofectamine2000 and A1AT-mRNA with the highest ratio. Statistical analysis suggests that there was a significant difference in secreted A1AT from the nebulized and non-nebulized fraction of lipoplexes prepared in this condition. The nebulized samples were not as efficient in transfecting 16HBE cells as the non-nebulized counterparts. However, the amount of secreted A1AT from the nebulized fraction with the highest Lipofectamine2000-to-A1AT-mRNA ratio was comparable to counterparts with the lowest and medium ratios. Collectively, these data indicate that both the nebulized and non-nebulized A1AT-mRNA-encoded A1AT protein (at least partially) are folded and modified appropriately within the airway epithelium to enable secretion out of the endoplasmic reticulum and into cell culture medium, thus the A1AT protein could be successfully detected in the supernatant.

### 3.4. Functional Test of the Secreted A1AT

After confirming the presence of the A1AT product in the supernatants of cells transfected by nebulized A1AT-mRNA/Lipofectamine2000 complexes, it is important to further verify whether the secreted protein was functional. A1AT is a general serine protease inhibitor; it is not only an inhibitor of the trypsin, but also can inhibit many other serine proteases, e.g., elastase. To that end, two functional assays were performed. We first evaluated the function of secreted A1AT in inhibiting the activity of trypsin. Trypsin is a serine protease that can hydrolyze the chromogenic substrate, Na-Benzoyl-DL-arginine-b-naphthylamide hydrochloride. The inhibition of the trypsin activity, e.g., by A1AT, prevents this reaction from occuring. The trypsin activity inhibited by secreted A1AT from supernatants of “nebulized” or “non-nebulized” A1AT-mRNA/Lipofectamine2000 complex-transfected cells was measured. Lipofectamine2000-treated counterparts and non-related mRNA (MetLuc-mRNA/Lipofectamine2000)-treated counterparts were added as mock and negative controls (NC). Both of these groups did not show significant enzyme inhibitory activity compared with the blank controls. Trypsin activity was maximally inhibited by an extent of 60% in samples from “non-nebulized lipoplexes with the highest Lipofectamine2000-to-A1AT-mRNA ratios” (Figure 9A), which is related to the amounts of detected A1AT in the supernatants. A1AT from cells transfected by “nebulized” lipoplexes could inhibit 36–48% of the trypsin activity, and the inhibition rates among samples prepared in different conditions were similar (Figure 9A).

The primary function of A1AT is to inhibit the activity of elastase. In order to evaluate the function of secreted A1AT in the supernatants of cells transfected by A1AT-mRNA lipoplexes, we also performed an anti-elastase assay in which elastase and its synthetic substrate (*N*-Methoxysuccinyl-Ala-Ala-Pro-Val-*p*-nitroanilide) were used to determine the extent of inhibition caused by A1AT [39]. The rate of enzymatic hydrolysis of the substrate is followed by the increase in absorbance due to the release of free *p*-nitroanilide cleaved from the substrate. Similar to the results of trypsin assay, samples from blank, mock and NC groups could not inhibit the activity of elastase. Samples from cells transfected by “nebulized” A1AT-mRNA lipoplexes could inhibit the elastase activity, and the inhibition rates (~35%) were comparable among samples prepared in different conditions (Figure 9B). A1AT presented in samples from “non-nebulized” A1AT-mRNA lipoplexes inhibited the elastase activity at more than 41% compared with the control, with a maximally inhibition rate of 54% in samples prepared with 10.8 µL Lipofectamine2000 (Figure 9B).

## 4. Discussion

Establishing safe, efficient and reproducible nebulized IVT-mRNA formulations will become the basis of successful gene therapy for various lung-related and respiratory diseases, such as cystic fibrosis and AATD. However, challenges remain in optimizing IVT-mRNA delivery to the lung [18,19], and there are yet extremely few studies focusing on the nebulized IVT-mRNA formulations designed for AATD treatment. As a result, we set out to investigate the optimal conditions for the preparation of A1AT-mRNA/Lipofectamine2000 complexes that can produce sufficient A1AT after the nebulization process.

The human bronchial epithelial cell line used in the current study, 16HBE14o- (16HBE), is widely used as a model for respiratory epithelial diseases and pulmonary gene therapy, due to the fact that 16HBE cells retain many of the functions and morphology of differentiated bronchial epithelial cells [40]. 16HBE cells are more static compared with other fast-dividing human bronchial epithelial cells such as BEAS-2B cells, thus being more difficult to transfect and better mimicking in vivo status [41]. Since Lipofectamine2000 has been demonstrated in several studies as an efficient transfecting agent in cultured cells [38,42,43], we employed this cationic lipid carrier throughout the current study. The medium in which complexes are prepared and incubated with cells poses an impact on transfection efficiency. We found the transfection efficiencies mediated by complexes prepared in glucose, sucrose or sodium chloride were much lower compared with the case of OptiMEM. Differences in the composition of these media may influence the condition of cultured cells, since OptiMEM and Ham’s F-12K contain more nutritious ingredients (such as amino acids, vitamins, hypoxanthine, thymidine, etc.) compared with glucose, sucrose or sodium chloride-based solutions; these ingredients would be helpful in keeping the cells in a better status for transfection. After the nebulization, an increase in the particle size, giving rise to a form of instability of the IVT-mRNA/Lipofectamine2000, could be observed in all the settings. However, the complexes remain in a nanometer size range following nebulization, which may still enable them to cross biological membranes and deliver their cargo into the cultured cells [44]. When triple amounts of IVT-mRNA were used to prepare the lipoplexes, the nebulized MetLuc-mRNA lipoplexes showed significantly higher transfection efficiency and considerably enhanced cytotoxicity. The increased initial amount of IVT-mRNA could ensure that more IVT-mRNA molecules survive the nebulization process. There will be more functional IVT-mRNA molecules in the nebulized fractions in this case, and the presence of more Lipofectamine2000 may help to alleviate the shear force during nebulization but also increase cytotoxicity in a dose-dependent manner [23,45]. We and others have demonstrated in previous studies that polyplexes prepared by cationic polymers such as branched or linear polyethylenimine could protect IVT-mRNA during the nebulization process, and their transfection efficiency was not significantly compromised after nebulization [23,46,47]. However, the transfection efficiency achieved by IVT-mRNA complexed with cationic lipids was much higher than counterparts formulated with cationic polymers in general, regardless of non-nebulized or nebulized formulation, as proven by previous studies [23,38].

In the current study, our main focus was to investigate the transfection efficiency of nebulized lipoplex formulation containing A1AT-mRNA, and to confirm that the secreted A1AT protein was still functional. We chose PARI Boy^®^ jet-nebulizer as the test setup, based on our previous experience [23,48]. Jet nebulizers have traditionally been used for the treatment of pulmonary diseases and are effective in delivering liposomal formulations [49]. Jet nebulizers are much cheaper compared with ultrasonic and mesh nebulizers, which would be helpful for a wider distribution and application of the aerosol technology [49]. Consistent with the results discussed above, the abundant A1AT production was observed in 16HBE cells transfected by A1AT-mRNA/Lipofectamine2000 complexes, before or after nebulization. The primary function of A1AT is to inhibit the activity of neutrophil elastase [50]. Elastase plays a significant role in the human immune system, since neutrophil granulocytes secrete it against Gram-negative bacteria [51]. However, when elastase secretion is out of control, the overexpressed elastase is able to cleave components of the extracellular matrix such as elastin, a macromolecule that provides the elastic recoil of the lung [52]. To inhibit neutrophil elastase, the active site amino acids Met 358–Ser 359 within A1AT will form a non-covalent interaction with the reactive site pocket of the elastase. Under a normal condition, this is a so-called suicide interaction for both proteins [26]. The results obtained from functional analysis implied that the secreted A1AT produced from both the nebulized and non-nebulized formulations could be enzymatically active in inhibiting the activity of trypsin as well as elastase. However, there was a significant decrease in the amount of secreted A1AT from the nebulized A1AT-mRNA lipoplexes compared to the non-nebulized fraction. This could be one indication that Lipofectamine2000 could not well protect A1AT-mRNA during the nebulization process. Further improvements could be done by employing the vibrating mesh nebulizer by which moderate shear forces are generated during nebulization, instead of the jet nebulizer whose shear forces could not be resisted by IVT-mRNA lipoplexes in the current study. Nevertheless, Lipofectamine2000-based formulations may still encounter limitations and challenges in the process of further development, considering its well-known in vivo toxicity, and they may not be effective when applied for in vivo purposes as the way they worked in cultured cells. Moreover, the most efficient IVT-mRNA/Lipofectamine2000 formulations in the current study were prepared in OptiMEM; it is probably not clinically practical to use the cell culture medium as the formulation solution. As a result, further works to improve the potency of nebulized A1AT-mRNA formulations are still required. For example, an alternative approach to improve the potency of A1AT-mRNA-based therapy would be the utilization of novel biomaterials that are specifically developed for nebulized IVT-mRNA formulations, an approach which displayed promising results in animal applications [24]. These delivery systems with good biocompatibility may pave the road for the development of aerosolized IVT-mRNA formulations, enabling safe and productive transfection in patients’ airways, to be applied to the successful treatment of AATD.

## 5. Conclusions

In summary, we first screened and improved the composition and transfection medium of IVT-mRNA/Lipofectamine2000 complexes. Their transfection efficiency was confirmed by measuring the amount of secreted protein and by evaluating the number of transfected cells. When the IVT-mRNA/Lipofectamine2000 complexes were prepared in OptiMEM, the lipoplexes displayed efficient transfection irrespective of the amount of IVT-mRNA (2 μL to 6 μL) or Lipofectamine2000 (1.2 μL to 4.8 μL) incorporated. IVT-mRNA lipoplexes enabled successful expression of A1AT in 16HBE cells without causing significant toxicity on the transfected cells. The nebulization process showed detrimental effects on the transfection potency of the IVT-mRNA lipoplexes, but these negative effects could be partially solved by increasing the components’ concentration within the IVT-mRNA lipoplexes (a triple amount would be sufficient, i.e., 18 μL IVT-mRNA within the lipoplex). 16HBE cells that were transfected by nebulized AIAT-mRNA formulations produced sufficient amounts of A1AT, even at the lowest Lipofectamine2000-to-A1AT-mRNA ratio (1:2.5), and the secreted A1AT could successfully inhibit the activity of trypsin and elastase in vitro. However, Lipofectamine2000 is limited for in vivo application; given its toxicity, alternative delivery systems that display promising in vivo profiles are necessary for the clinical translation process. Together, these findings suggest that IVT-mRNA-encoding A1AT was still functional in nebulized formulations, which can be further developed in the future as an attractive route of administration for clinical applications within the field of pulmonary-based A1AT treatments.

## Figures and Tables

**Figure 1 pharmaceutics-13-01281-f001:**
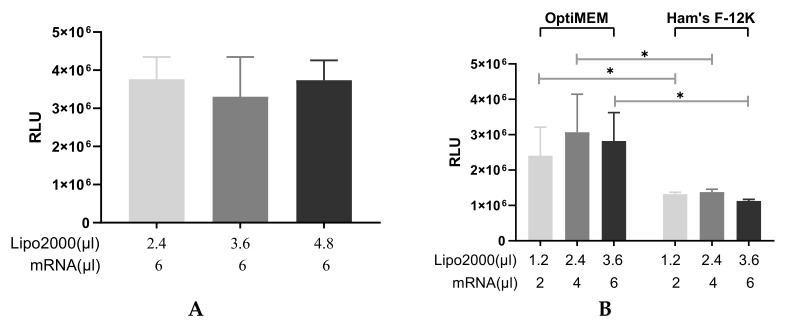
Optimization and characterization of the transfection process mediated by different IVT-mRNA/Lipofectamine2000 complexes. (**A**) Transfection efficiency of complexes prepared at different Lipofectamine2000-to-IVT-mRNA ratios in OptiMEM. Totals of 2.4 µL, 3.6 µL or 4.8 µL Lipofectamine2000 were mixed with 6 µL IVT-mRNA-encoding Metridia luciferase (0.1 µg/µL), n = 4. (**B**) Transfection efficiency of complexes prepared in OptiMEM or Ham’s F-12K medium, n = 4. (**C**) Transfection efficiency of MetLuc-mRNA lipoplexes prepared in 0.9% sodium chloride solution, n = 6. (**D**) Transfection efficiency of complexes incubated in glucose and sucrose solution. The complexes were prepared in saline, then incubated with 16HBE cells in 5% glucose or 5% sucrose solution, n = 6. “Lipo2000” represents Lipofectamine2000 and “mRNA” represents MetLuc-mRNA. All complexes were incubated with 16HBE cells for 2 h. The transfection efficiency was evaluated 24 h after adding complexes to the cells. The results are presented as relative light units (RLU). A two-tailed Student’s *t*-test was used to determine significance (* *p* < 0.05).

**Figure 2 pharmaceutics-13-01281-f002:**
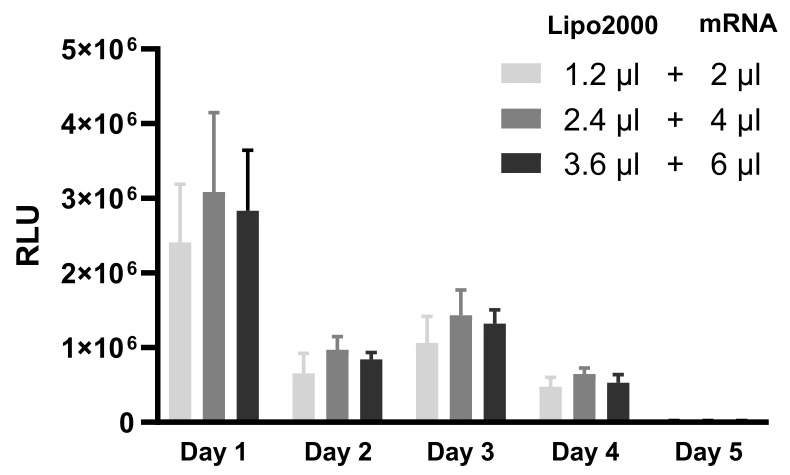
Time course of luciferase secretion after transfection with IVT-mRNA lipoplexes. Complexes were prepared by mixing 1.2 µL, 2.4 µL or 3.6 µL Lipofectamine2000 with 2 µL, 4 µL or 6 µL MetLuc-mRNA in OptiMEM. “Lipo2000” represents Lipofectamine2000 and “mRNA” represents MetLuc-mRNA. 16HBE cells were incubated with the complexes for 2 h. The luciferase activity was measured every 24 h till the relative light units measured with a luminometer dropped below 500 (background level). After each measurement, the medium was removed and the cells were supplied with fresh cell culture medium. The results are presented as relative light units (RLU), n = 4.

**Figure 3 pharmaceutics-13-01281-f003:**
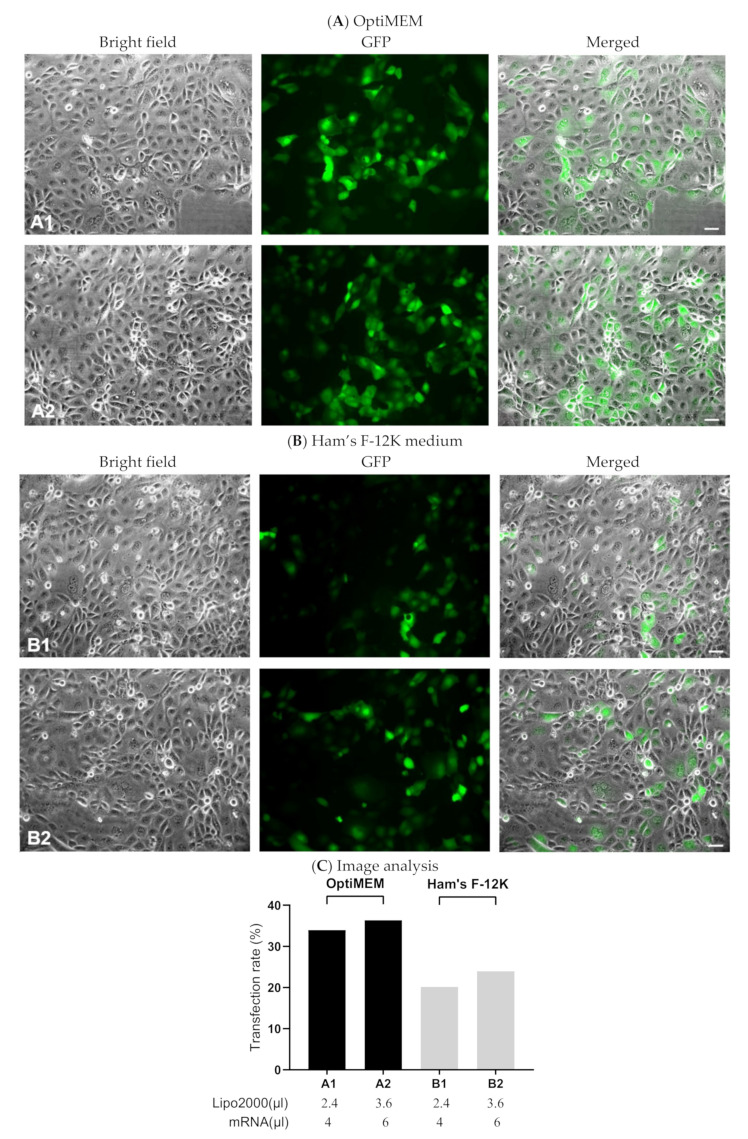
Transfection efficiency of GFP-mRNA on 16HBE cells. Totals of 2.4 µL or 3.6 µL Lipofectamine2000 were mixed, respectively, with 4 µL or 6 µL IVT-mRNA-encoding GFP (0.1 µg/µL). Complexes were prepared in OptiMEM (**A**) or Ham’s F-12K medium (**B**). 16HBE cells were incubated with GFP-mRNA/Lipofectamine2000 complexes for 2 h. The pictures were taken 24 h after transfection with an Axiovert fluorescence microscope. Scale bar: 40 μm. (**C**) Quantitative analysis on the rates of successfully transfected cells in the images of (**A**,**B**). “Lipo2000” represents Lipofectamine2000 and “mRNA” represents GFP-mRNA.

**Figure 4 pharmaceutics-13-01281-f004:**
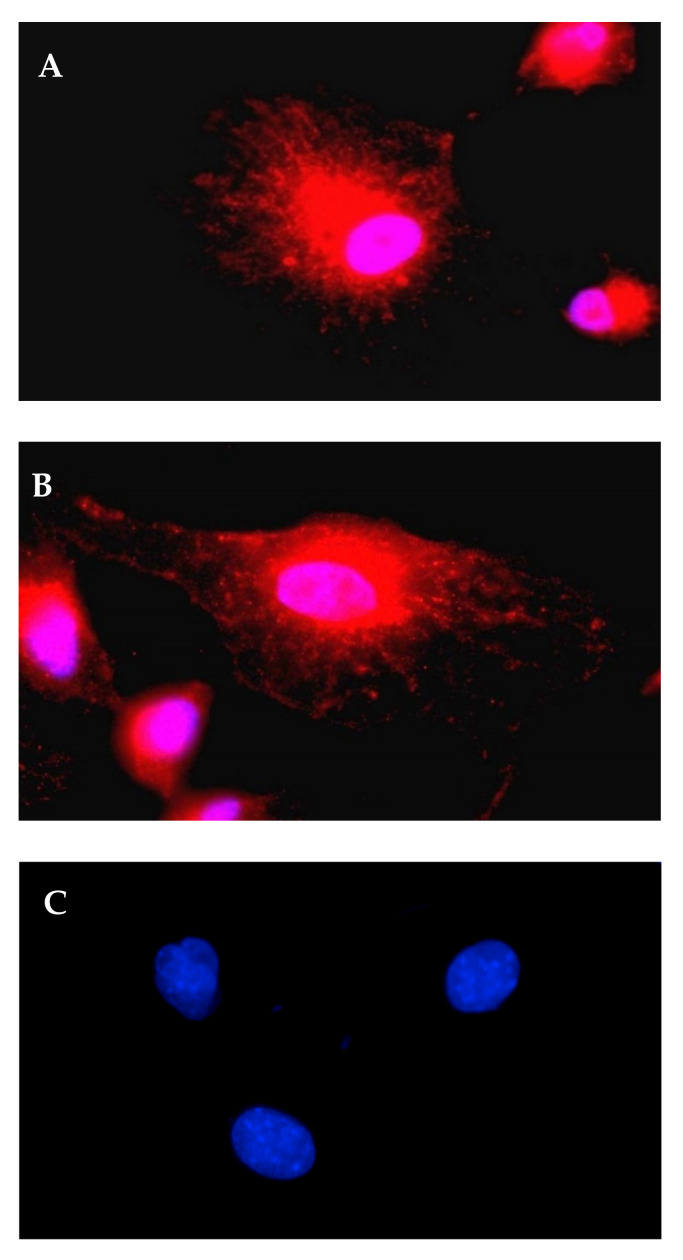
Intracellular A1AT visualized by immunofluorescence. (**A**,**B**) are representative pictures of A1AT-mRNA-transfected cells obtained from different fields. (**C**) Control of non-transfected cells. A total of 4.8 µL Lipofectamine2000 was mixed with 6 µL A1AT-mRNA (0.1 µg/µL) in OptiMEM. 16HBE cells were incubated with the complexes for 2 h. Brefeldin-A was used to prevent the protein secretion. Pictures were taken 24 h after transfection using an Axiovert fluorescence microscope with 100× magnification. Red fluorescence represents A1AT; blue or pink areas were cell nuclei-stained by DAPI.

**Figure 5 pharmaceutics-13-01281-f005:**
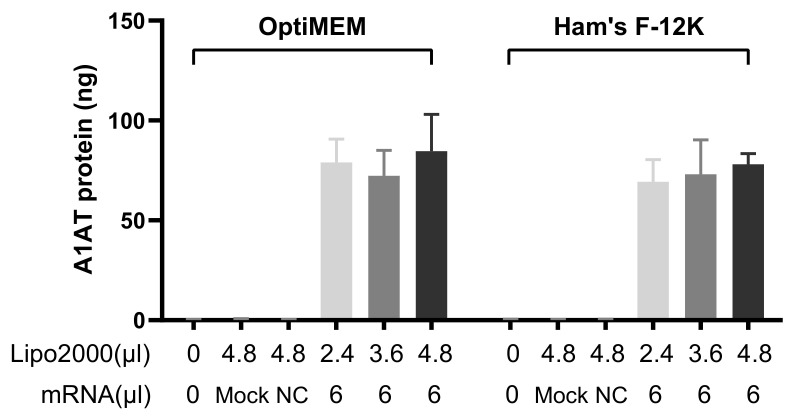
Transfection efficiency mediated by A1AT-mRNA/Lipofectamine2000 complexes. Detected amounts of secreted A1AT after transfection with IVT-mRNA/Lipofectamine2000 complexes using an ELISA assay. Totals of 2.4 µL, 3.6 µL or 4.8 µL Lipofectamine2000 were mixed with 6 µL A1AT-mRNA. Untreated samples were used as blank controls, 4.8 µL Lipofectamine2000-treated samples served as “Mock” controls and lipoplexes prepared by 6 µL MetLuc-mRNA and 4.8 µL Lipofectamine2000 were applied as negative controls (“NC”). “Lipo2000” represents Lipofectamine2000 and “mRNA” represents A1AT-mRNA (except Mock and NC). OptiMEM and Ham’s F-12K medium were used to prepare complexes. 16HBE cells were incubated with the complexes for 2 h. To calculate the amounts of produced and secreted A1AT, a standard curve was employed; n = 4.

**Figure 6 pharmaceutics-13-01281-f006:**
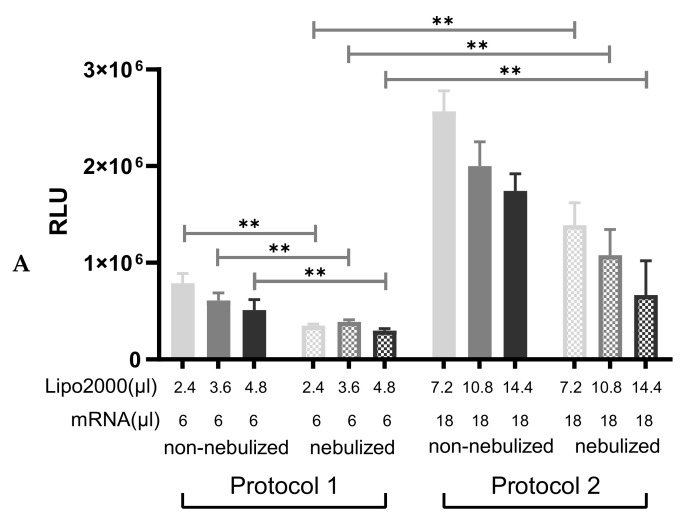
Transfection efficiency of MetLuc-mRNA/Lipofectamine2000 complexes before and after nebulization via different nebulization protocols. Lipoplexes were prepared in OptiMEM (**A**) or Ham’s F-12K medium (**B**). For protocol 1, 2.4 µL, 3.6 µL and 4.8 µL Lipofectamine2000 was mixed with 6 µL MetLuc-mRNA (0.1 µg/µL). After 10 min, the samples were diluted in 3.0 mL medium. For protocol 2, complexes were prepared with 7.2 µL, 10.8 µL and 14.4 µL Lipofectamine2000 mixed with 18 µL MetLuc-mRNA (0.1 µg/µL). “Lipo2000” represents Lipofectamine2000 and “mRNA” represents MetLuc-mRNA. After 10 min, the samples were diluted in 3.0 mL medium. For both protocols, a fraction of the complexes was kept separately and used as a “non-nebulized” control (solid bars). The rest of the solution was aerosolized for 5 min using a PARI Boy^®^ jet-nebulizer and the collected part was used as “nebulized” formulation (dots filled bars). Both the non-nebulized control complexes and the nebulized complexes were used to transfect 16HBE cells for 2 h. The transfection efficiency was measured 24 h after transfection. The results are presented as relative light units (RLU), n = 5. Significance was determined by a two-tailed unpaired Student’s *t*-test (* *p* < 0.05 and ** *p* < 0.01).

**Figure 7 pharmaceutics-13-01281-f007:**
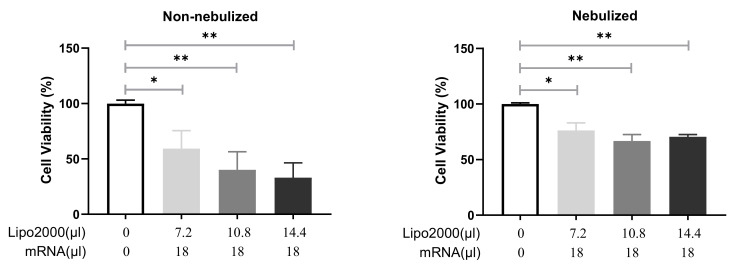
Cytotoxicity profile of MetLuc-mRNA/Lipofectamine2000 complexes prepared by Protocol 2 towards 16HBE cells. The complexes were prepared in OptiMEM and incubated with 16HBE cells for 2 h. Totals of 7.2 µL, 10.8 µL and 14.4 µL Lipofectamine2000 were mixed with 18 µL MetLuc-mRNA. “Lipo2000” represents Lipofectamine2000 and “mRNA” represents MetLuc-mRNA. A fraction of the complexes was kept separately and used as a “non-nebulized” sample. The rest of the solution was aerosolized for 5 min using a PARI Boy^®^ jet-nebulizer and the collected part was used as a “nebulized” formulation. Cell viability was assayed 24 h after transfection. Untreated cells were used as 100%, n = 6. The results were analyzed for the statistical significance with a two-tailed unpaired Student’s *t*-test: * *p* < 0.05 and ** *p* < 0.01.

**Figure 8 pharmaceutics-13-01281-f008:**
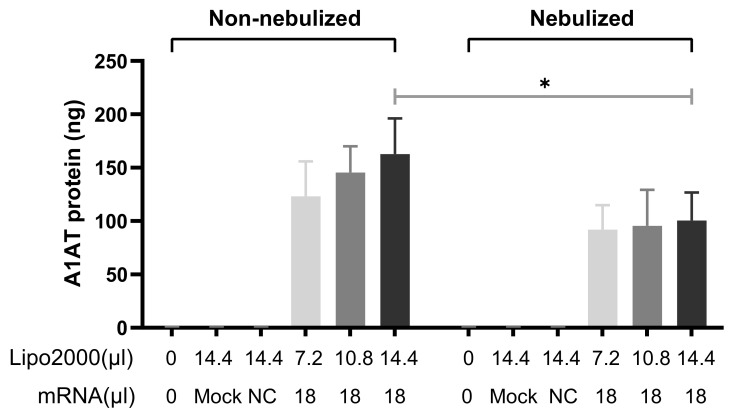
Nanograms of secreted A1AT after transfection with nebulized and non-nebulized A1AT-mRNA/Lipofectamine2000 complexes. Totals of 7.2 µL, 10.8 µL or 14.4 µL Lipofectamine2000 and 18 µL IVT-mRNA-encoding A1AT (0.1 µg/µL) were mixed in order to prepare complexes. Untreated samples were used as blank controls. A total of 14.4 µL Lipofectamine2000-treated samples served as a “Mock” control. Lipoplexes prepared by 18 µL MetLuc-mRNA and 14.4 µL Lipofectamine2000 were applied as negative controls (NC). “Lipo2000” represents Lipofectamine2000 and “mRNA” represents A1AT-mRNA (except Mock and NC). OptiMEM was used as the transfection medium. A fraction of the complexes was kept separately and used as a “non-nebulized” control. The other fraction was aerosolized using a PARI Boy^®^ jet-nebulizer and the “nebulized” formulation was collected. Both the non-nebulized complexes and the nebulized complexes were used to transfect 16HBE cells for 2 h. The amount of secreted A1AT in supernatants was evaluated 24 h after transfection. A standard curve was employed for the calculation, n = 4. Significance was determined by two-tailed unpaired Student’s *t*-test (* *p* < 0.05).

**Figure 9 pharmaceutics-13-01281-f009:**
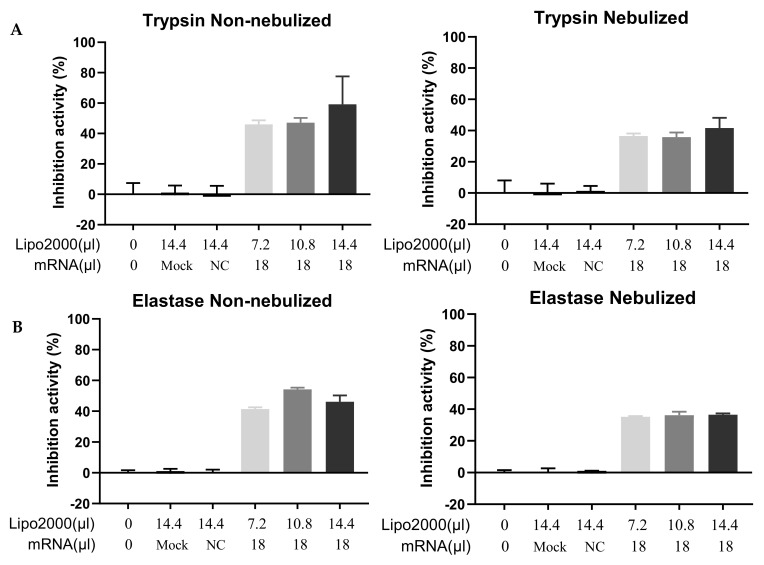
Functional assay of A1AT in supernatants of transfected cells. (**A**) Inhibition of trypsin activity by A1AT secreted from 16HBE cells transfected by non-nebulized or nebulized A1AT-mRNA/Lipofectamine2000 complexes, n = 8. (**B**) Inhibition of elastase activity by A1AT secreted from 16HBE cells transfected by non-nebulized or nebulized A1AT-mRNA lipoplexes, n = 4. For both assays, 7.2 µL, 10.8 µL or 14.4 µL Lipofectamine2000 and 18 µL IVT-mRNA-encoding A1AT (0.1 µg/µL) were mixed to prepare complexes. A total of 14.4 µL Lipofectamine2000-treated samples served as “Mock” controls. A total of 18 µL MetLuc-mRNA was formulated with 14.4 µL Lipofectamine2000, which was applied as negative control “NC”. “Lipo2000” represents Lipofectamine2000 and “mRNA” represents A1AT-mRNA (except Mock and NC). 16HBE cells were incubated with the A1AT-mRNA lipoplexes for 2 h and the medium was then replaced by a protein expression medium. The enzyme inhibition assay was performed 24 h after transfection. A total of 50 µL cell culture supernatants was added to the reaction buffer, and the enzyme inhibition was assessed after the reaction. Untreated samples were used as blank controls and their absorbance were set as 100% (whose inhibition activity was 0%, correspondingly).

**Table 1 pharmaceutics-13-01281-t001:** Size of IVT-mRNA/Lipofectamine2000 lipoplexes.

IVT-mRNA/Lipofectamine2000 Lipoplexes	Amounts of Lipofectamine2000 (µL)	Amounts of IVT-mRNA (µL)	Size (nm) Non-Nebulized	Size (nm) Nebulized
1st Setting	2.4	6.0	397 ± 118	649 ± 173
1st Setting	3.6	6.0	467 ± 67	543 ± 123
1st Setting	4.8	6.0	324 ± 46	628 ± 76
2nd Setting	1.2	2.0	458 ± 89	737 ± 113
2nd Setting	2.4	4.0	480 ± 74	676 ± 138
2nd Setting	3.6	6.0	378 ± 84	586 ± 161

The IVT-mRNA lipoplexes were prepared by mixing different amounts Lipofectamine2000 with IVT-mRNA. The complexes were prepared in OptiMEM. After a 10 min incubation period, the size measurement was performed. The data represent hydrodynamic diameter ± SD, n = 3.

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
