# Peer review of "In Vitro Investigations on Optimizing and Nebulization of IVT-mRNA Formulations for Potential Pulmonary-Based Alpha-1-Antitrypsin Deficiency Treatment†"

_pharmaceutics, 2021, doi:10.3390/pharmaceutics13081281_

Round 1

Reviewer 1 Report

Only minor points about text to change in revised manuscript

1) To follow the description explaining cell name above and below, it seems to be better to change in Line 288-289 ~ human bronchial epithelial cells (16HBE14o-, 16HBE).

→ human bronchial epithelial cells [16HBE14o- (16HBE)].

2) To keep universality,

in Figure 3 graph title

Opti-MEM → OptiMEM

in Figure 9 graph X-axis

MOCK → Mock

Reviewer 2 Report

The authors did a thorough revision of the manuscript, which clearly improved it in quality. In particular, the figures are much easier to understand as “stand alone” items in the revised version. In addition, important controls were added.

Before publishing, there are however still some minor issues:

Figure 3: y-axis should be transfection rate (%)

Also in Fig 3: a size bar in micrographs is standard.

Some English editing should be performed. For example “samples did not treat” could be "untreated"

In the conclusion: I agree that Lipofectamine is no viable option for in vivo application. Still, having performed an in vitro optimization study for transfection, I would consider it adequate that the conclusion could summarize the found optimum transfection conditions under used settings.

Author Response

This manuscript is a resubmission of an earlier submission. The following is a list of the peer review reports and author responses from that submission.

Round 1

Reviewer 1 Report

Guan et al try to establish delivery system for IVT-mRNA with nebulization to bronchial epithelial cells in vitro. Authors show that nebulized IVT-mRNA after delivery through lipoplexes (using Lipofectamine 2000), can produce A1AT protein and secrete functional enzyme from bronchial cells, at least in vitro.

Major points

1. In Figure 3, to conclude less GFP positive cells in Ham’s F-12K medium samples compare to that of OptiMem, some image analyses should be required.

2. Some controls are required in Figure 8 and 10 to conclude.

In Figure 8, cell viability of non-nebulized RNA transfectants. (Otherwise, we do not know whether the difference of transfection efficiency in Figure 7 and amount of A1AT protein in Figure 9 containing factor of viability after transfection.)

In Figure 10, the effect of supernatant from transfectants with mock (at least), mutant A1AT-mRNA (this is the best control), or non-related-mRNA (GFP, MetLuc, etc) for trypsin and elastase activities as a negative control.

Minor points

1. In Materials and Methods section

~ Transfection of Cultured Cells

Please explain in detail what is Naked IVT-mRNA

~ Nebulization

“The complexes were incubated with the cells for 2h.”

It is not clear whether nebulized complex mix with cells, without or with    medium.

“After their removal 0.1 ml of fresh culture medium ~”

→ After their removal, 0.1ml of fresh culture medium ~

2. In Figure 4, authors should explain this is two different fields (and representative)

3. Possibly, this is just PDF format problem, but in Introduction section,

           TNFα (alpha ~ different font)

           IL-1b (beta ~ not b)

Similarly, please check simple mistake

   Line 174  OptiMEM → OptiMem

   Line 238  medium.50 uL → medium. 50 uL

   Line 290  IVT-mRNA/Lipofectamine200 → IVT-mRNA/Lipofectamine2000

4. Text shift/move to weird position in several Figures.

   C and D in Figure 1

   Condition 1 and condition 2 in Figure 3

   A and B in Figure 5

5. In Figure10A, graph for Nebulized seems to be wrong position and part of Y axis is covered by text.

Reviewer 2 Report

The manuscript reports a study on transfection of an airway cell line by Lipofectamin enabled mRNA delivery. The goal was to optimize the in vitro transfection protocol and to test suitability for aerosolization.

In general: The manuscript in its current form is not suitable for publication in pharmaceutics. Most important the manuscript would profit from shortening and better organization of the presented results. An article aiming for optimization needs to sort conditions, decide, and present some optimum instead of comparing the same conditions throughout the full manuscript (always 3 concentrations and 2 different media). The naming of the samples as condition 1, 2 and 3 and setting 1,2, and 3 is also very difficult for the reader. Using the mRNA amount or mRNA:Lipofectamin ratio directly would be more straight forward. The bar charts should always use the same color for the same sample. Bar charts with different grey shades and with more different colors than sample number without color legend are difficult.

The introduction section is too long. Shorter section about pulmonary mRNA therapy and alpha-1-antitrypsin therapy would be sufficient. Instead some rational for the selection of the in vitro model would be needed. Why the authors decides for 16HBE cells and Pariboy jet nebulizer as test set-up?

Material+Methods section:

“stabilized non-immunogenic mRNA” need to be better explained. Is that realized by some chemical modification?

Please give the full name of 16HBE cells (16HBE14o-).

Were cell confluent or sub-confluent at the time of transfection? This info is crucial since the confluency has impact on transfection efficacy and cytotoxicity.

MTT assay usually is performed directly after treatment/incubation. Fast growing cells could compensate for some cell death when the assay was performed after 24h incubation, leading to some underestimation of cytotoxicity.

Controls for alpha-1-antitrypsin expression of 16HBE cells are needed as non treated and mock(vehicle) treated sample to ensure the measured protein all origins from the delivered mRNA.

Results

Figure1: Use same y-axis (x106) for all 4 diagrams.

Figure 3: the micrographs are not separated, thus it is unclear where one ends and the second starts. The labels are not correctly placed. An overlay of light and fluorescence picture would provide a better estimate of the ratio of transfected vs non-transfected cells.

Figure 4: Control of non-transfected 16HBE cells needed. Low cell number without overview picture on cells is not very informative.

Figure 5: Photo of the ELISA is not required since it does not provide information

Figure 6: remove or shift to supplemental information and describe as text – since 6 graphs without differences are using unnecessary space.

Figure 10: Could help to bring easier the essential message by calculation of the “inhibition activity”?

The discussion part is for the larger part just a repetition of the results and not discussion further aspects (e.g. Differences in the composition of the used media and implications for pulmonary application? What would be the advantages of a nebulized mRNA compared to nebulized protein therapy? )

Did the authors measure the size of the 3-fold concentrated lipofectamine formulation?

If the authors hypothesize that jet nebulization damages the mRNA, why was that hypothesis not checked by a gel electrophoresis?

Superior in vitro test system would use air-interface deposition of the aerosol on a differentiated epithelium, which could lead to different transfection. On the one hand, direct deposition (without non-physiologic liquid layer dimensions) can increase uptake. At the other hand, fully differentiated cells are harder to transfect.